# A Single-Peak-Structured Solar Cycle Signal in Stratospheric Ozone based on Microwave Limb Sounder Observations and Model Simulations

Sandip S. Dhomse[1,2], Martyn P. Chipperfield[1,2], Wuhu Feng[1,3], Ryan Hossaini[4], Graham W. Mann[1], Michelle L. Santee[5], and Mark Weber[6]

[1]School of Earth and Environment, University of Leeds, Leeds, UK
[2]National Centre for Earth Observation, University of Leeds, Leeds, UK
[3]National Centre for Atmospheric Science, University of Leeds, Leeds, UK
[4]Lancaster Environment Centre, Lancaster University, Lancaster, UK
[5]Jet Propulsion Laboratory, California Institute of Technology, Pasadena, CA, USA
[6]Institute of Environmental Physics, University of Bremen, PO Box 330 440, D-28334 Bremen, Germany

**Correspondence:** Sandip S. Dhomse (s.s.dhomse@leeds.ac.uk)

**Abstract.** Until now our understanding of the 11-year solar cycle signal (SCS) in stratospheric ozone has been largely based on high quality but sparse ozone profiles from the Stratospheric Aerosol and Gas Experiment (SAGE) II or coarsely resolved ozone profiles from the nadir-viewing Solar Backscatter Ultraviolet Radiometer (SBUV) satellite instruments. Here, we analyse 16 years (2005-2020) of ozone profile measurements from the Microwave Limb Sounder (MLS) instrument on the Aura satellite to estimate the 11-year SCS in stratospheric ozone. Our analysis of Aura-MLS data suggests a single-peak-structured SCS profile (about 3% near 4 hPa or 40 km) in tropical stratospheric ozone, which is significantly different to the SAGE II and SBUV-based double-peak-structured SCS. We also find that MLS-observed ozone variations are more consistent with ozone from our control model simulation that uses Naval Research Laboratory (NRL) v2 solar fluxes. However, in the lowermost stratosphere modelled ozone shows a negligible SCS compared to about 1% in Aura-MLS data. An ensemble of Ordinary Least Square (OLS) and three regularised (Lasso, Ridge and ElasticNet) linear regression models confirms the robustness of the estimated SCS. In addition, our analysis of MLS and model simulations shows a large SCS in the Antarctic lower stratosphere that was not seen in earlier studies. We also analyse chemical transport model simulations with alternative solar flux data. We find that in the upper (and middle) stratosphere the model simulation with Solar Radiation and Climate Experiment (SORCE) satellite solar fluxes is also consistent with the MLS-derived SCS and agrees well with the control simulation and one which uses Spectral and Total Irradiance Reconstructions (SATIRE) solar fluxes. Hence, our model simulation suggests that with recent adjustments and corrections, SORCE data can be used to analyse effects of solar flux variations. Furthermore, analysis of a simulation with fixed solar fluxes and one with fixed (annually repeating) meteorology confirms that the implicit dynamical SCS in the (re)analysis data used to force the model is not enough to simulate the observed SCS in the middle and upper stratospheric ozone. Finally, we argue that the overall significantly different SCS compared to previous estimates might be due to a combination of different factors such as much denser MLS measurements, almost linear stratospheric chlorine loading

changes over the analysis period, variations in the stratospheric dynamics as well as relatively unperturbed stratospheric aerosol layer that might have influenced earlier analyses.

*Copyright statement.* TEXT

# 1 Introduction

Changes in solar irradiance over the 11-year cycle are an important external forcing to the climate system. As the largest changes occur at shorter wavelengths, such as the ultra-violet (UV) part of the solar spectrum, detecting related changes in stratospheric ozone is an obvious approach to improve our understanding of solar–climate interactions (e.g. Gray et al., 2010). Increased UV radiation during solar maximum enhances photolysis of oxygen at shorter UV wavelengths leading to ozone production, while at longer UV wavelengths enhanced ozone photolysis leads to net ozone loss through increased 30 concentrations of atomic oxygen (Haigh, 1994).

Though many chemical models have suggested a single-peak-structured solar cycle signal (SCS) in stratospheric ozone (e.g. SPARC, 2010, Chap. 10), observation-based estimates differ widely. Chandra (1984) performed an initial attempt to estimate SCS using satellite-derived stratospheric ozone profiles from Nimbus-4 Backscatter Ultra-Violet (BUV) radiometer data for the 1970-1976 time period. Their analysis suggested up to 12% decrease in upper stratospheric ozone from solar maximum to solar 35 minimum. Later, Hood (1993) analysed 11.5 years (January 1979 to June 1990) of Nimbus-7 Solar BUV (SBUV) data and suggested that the upper stratospheric SCS is significantly smaller than the earlier estimate (about 8%). Chandra and McPeters (1994), Fleming et al. (1995) and McCormack and Hood (1996) also analysed about 15 years (1979-1993) of SBUV data to report a SCS of about 6-8% near 2 hPa, and a minimum response in the mid-stratosphere. Similarly, Chandra et al. (1996) found that upper stratosphere ozone profiles from the Microwave Limb Sounder (MLS) on-board the Upper Atmospheric 40 Research Satellite (UARS) displayed a similar magnitude of ozone change, that is about 5% UV decrease (averaged between 200-205 nm) during the declining phase of solar cycle 22, which led to about 2-4% ozone decrease in the upper stratosphere. In contrast, Wang et al. (1996) analysed Stratospheric Aerosol and Gas Experiment (SAGE) I and SAGE II ozone profiles (1979–1991) to find an almost negligible SCS in the upper stratosphere.

With the successful implementation of the Montreal Protocol, some satellite data were able to detect decreases in the upper 45 stratospheric chlorine loading. Hence, some studies such as Newchurch et al. (2003) analysed SAGE I /II (1979–2003) and Halogen Occultation Experiment (HALOE, 1991–2003) data to suggest early signs of ozone recovery by the year 2000 in upper stratospheric ozone. However, Steinbrecht et al. (2004) analysed mid-latitude lidar-radar profiles (1987–2003) and argued that increased solar activity might have been responsible for a sudden increase in upper stratospheric ozone after the year 2000.

Later, Soukharev and Hood (2006) analysed 25 years of SBUV/SBUV2 (1979–2003) ozone profiles to show a minimum 50 SCS in the middle stratosphere and up to 2% SCS in the upper stratosphere. In contrast to Wang et al. (1996), their analysis of SAGE II data (1985–2003) showed up to 4% SCS in the upper stratosphere but HALOE (1992–2003) data indicated opposite

trends of about -2% SCS in the middle stratosphere. In contrast, Remsberg (2008) and Remsberg and Lingenfelser (2010) also analysed HALOE and SAGE II ozone profiles for the HALOE time period (1992-2005) to show first and second peaks near 32 km (5 hPa) and 50 km (0.5 hPa), respectively. Recently, Dhomse et al. (2016) and Maycock et al. (2016) analysed updated SAGE V7.0 ozone profiles to show a significantly reduced SCS in the upper stratosphere. Both of those studies also noted that the SCS structure is altered significantly if the analysis is performed in mixing ratio units rather than native number density units. Recently, Ball et al. (2019) analysed updated BAyeSian Integrated and Consolidated (BASIC V2) data (1984-2016) that also showed a double-peak-structured SCS with primary peak near 35 km and secondary peak near 24 km.

Though most of the observation-based studies suggested a double-peak-structured SCS, initial 2-D model studies (Garcia et al., 1984; Brasseur, 1993; Huang and Brasseur, 1993; Fleming et al., 1995) could simulate only a single-peak-structured SCS in the middle stratosphere. The lack of double-peak structure in the chemical models was attributed to discrepancies in the 2-D transport. Later, Dhomse et al. (2011) used a 3-D chemical transport model (CTM) to successfully simulate a double-peak-structured SCS over 1979–2005 time period. However, most free-running 3-D chemistry-climate models (CCMs) also simulate only a single-peak-structured SCS in the tropical middle stratosphere (see SPARC, 2010, Figure 8.11). The inability of CCMs to simulate a SBUV/SAGE-type SCS is generally attributed to inadequate or missing representation of key dynamical processes such as the Quasi-Biennial Oscillation (QBO), El Nino/Southern Oscillation, changes in the meridional circulation and stratospheric aerosol-induced chemical/dynamical changes following the El Chichon and Mt. Pinatubo volcanic eruptions (e.g. Lee and Smith, 2003; Smith and Matthes, 2008; Dhomse et al., 2011, 2015, 2020; Chiodo et al., 2014).

Another important factor has been large uncertainties in solar flux measurements (e.g. Ermolli et al., 2013). Most model simulations are forced with solar irradiance variability from (semi)empirical models such as NRL and SATIRE (e.g. Lean, 2000; Krivova et al., 2010; Yeo et al., 2014; Coddington et al., 2016) that are in general good agreement with many solar observations (Lean and DeLand, 2012; Coddington et al., 2019). However, with the launch of the Solar Radiation and Climate Experiment (SORCE) satellite in January 2003, high resolution solar irradiance measurements suggested significantly different UV variability (Harder et al., 2009). Using SORCE measurements some modelling studies (Haigh et al., 2010; Merkel et al., 2011; Swartz et al., 2012) suggested a negative SCS in the upper stratosphere/lower mesosphere (US/LM). These studies included analysis of few years of MLS and Sounding of the Atmosphere using Broadband Emission Radiometry (SABER) datasets to show consistent changes in the observed ozone profiles. In contrast, Dhomse et al. (2013) used the same SORCE fluxes and found that SORCE-based solar spectral irradiance (SSI) changes were not enough to explain observed ozone changes. Other studies soon confirmed that initial versions of SORCE data overestimated UV variability (e.g. Ermolli et al., 2013; Haberreiter et al., 2017).

An important aspect of solar flux variability has been differences in terms of sunspot numbers (SSN) and their durations over different solar cycles (e.g. Chapman et al., 2020). For example, SILSO World Data Center, 2021 data clearly shows significantly different maximum monthly SSNs during solar cycle 21 ($\approx$ 210), 22 ($\approx$ 200), 23 ($\approx$ 150) and 24 ($\approx$ 100). This clearly highlights that recent solar cycles had values about 200 reducing to 150 and 100 during solar cycles 23 and 24, respectively. This indicates that solar flux variability (solar maxima minus solar minima) would have different characteristics over different solar cycles. Hence, Dhomse et al. (2015) analysed model and satellite data sets over different time period to

show differences in SCS magnitudes depending on analysis period such as 1979–2013 (SBUV), 1984–2005 (SAGE), 1992–2005 (HALOE), 2004–2013 (MLS). However, for each analysis period, both satellite and model-simulated ozone profiles showed a double-peak-structured SCS in the tropical stratospheric ozone. It is important to note that the SBUV, SAGE II and
HALOE analysis periods include years where the stratospheric aerosol layer was strongly perturbed by El Chichon and/or Mt Pinatubo volcanic eruptions.

Overall, there is still a large uncertainty in our understanding of the true nature of the ozone SCS profile as most estimates rely on sparsely sampled solar occultation instruments (SAGE II, HALOE) or SBUV data with poor vertical resolution, and may depend on the time period considered (e.g. Remsberg and Lingenfelser, 2010; Dhomse et al., 2015) . Here, we analyse 16
95  years (2005–2020) of updated, high quality and densely sampled MLS ozone profiles to quantify the stratospheric SCS. We also use the TOMCAT/SLIMCAT 3-D CTM to analyse effects of different updated solar fluxes. Finally, we present the estimated SCS profile using different linear regression models such as Ordinary Least Square (OLS), Lasso, Ridge, and ElasticNet. The model setup and satellite data used here are described in Section 2 followed by details of our regression model in Section 3. Key results are discussed in Section 4.

## 2  Model Set Up and Satellite data

We have performed simulations with the TOMCAT three-dimensional CTM (Chipperfield, 2006; Chipperfield et al., 2017) for the 2004–2020 time period. The model setup is similar to the control simulation used in our recent studies (e.g. Dhomse et al., 2019; Feng et al., 2021; Weber et al., 2021). Briefly, the model contains a detailed description of stratospheric chemistry and is forced using European Centre for Medium-Range Weather Forecasts Fifth generation reanalysis (ERA-5) meteorological fields
(Hersbach et al., 2020). Model simulations are performed at $2.8° \times 2.8°$ horizontal resolution with 32 levels ranging from the surface to ∼60 km. Surface concentrations of ozone depleting substances (ODSs) and greenhouse gases are from Engel et al. (2018b). Stratospheric sulfate aerosol surface density (SAD) data are from ftp://iacftp.ethz.ch/pub_read/luo/CMIP6/ and updated since Dhomse et al. (2015) to extend until 2018. As the equivalent SAD values are not yet released for later years, we use monthly averaged SAD (1996–2005) for 2019 and 2020. Thus, our analysis will miss the impact on ozone of SAD changes
following the Raikoke and Ulawun eruptions in June 2019. The model also includes contributions from four chlorinated very short-lived substances ($CH_2Cl_2$, $CHCl_3$, $C_2Cl_4$, and $C_2H_4Cl_2$) as described in Hossaini et al. (2017, 2019). Additionally, the model includes a fixed 5 ppt of stratospheric $Br_y$ from brominated VSLS $CHBr_3$ (1 ppt) and $CH_2Br_2$ (1 ppt) (e.g. Feng et al., 2007).

To understand the effects of solar irradiance variability on the evolution of ozone, we performed five simulations with
different solar fluxes. Three simulations use solar irradiance variability from NRLSSI ~~V2~~ v2 (hearafter NRL2, Coddington et al., 2016), SATIRE (Yeo et al., 2014), SORCE (Harder et al., 2019) and are labelled **A_NRL**, **B_SAT** and **C_SOR**, respectively. As TOMCAT has 203 relatively coarse spectral bins in the photolysis scheme (Lyman alpha and 170-850 nm), daily high-resolution SSI data sets are integrated for the model spectral bins before calculating monthly means (e.g. Dhomse et al., 2011, 2013). To quantify the effect of any implicit SCS in ERA5 reanalysis data, we also performed a fourth model simulation,

**D_SFix**, which uses constant solar fluxes for the entire 2005–2020 time period. To separate chemical effects of time-varying solar fluxes, a fifth simulation (**E_DFix**) uses NRL2 solar fluxes but fixed dynamic forcing (annually repeating dynamical fields from year 2004). NRL2, SATIRE, and SORCE v19 data are obtained via the Laboratory for Atmospheric and Space Physics (LASP) Solar Irradiance data Center (https://lasp.colorado.edu/lisird/) at the University of Colorado.

This study primarily focuses on the analysis of MLS version 5 (v5) data. Daily MLS ozone profiles are obtained from https://disc.gsfc.nasa.gov/datasets?page=1&keywords=ML2O3_005 (last access : June 2021). MLS profiles have been filtered according to the guidelines specified by Livesey et al. (2020), who provide a critical analysis of the v5 dataset. Briefly, the scientifically useful altitude range for MLS ozone profiles is from 261 hPa to 0.001 hPa. The retrieval precision (∼2%) and accuracy (∼6%) are optimum near 10 hPa but degrade above and below that level, reaching values of 30% and 10%, respectively, at 0.2 hPa and 100 hPa, the extremes of the domain shown in this study. MLS zonal monthly means are calculated by binning the profiles onto 64 latitude intervals (TOMCAT model latitudes).

## 3   Multivariate Regression Model

Here we use an ensemble of multivariate linear regression (MLR) models to estimate the SCS in both MLS and TOMCAT ozone profiles. The basic MLR set up is a slightly modified version to that used in Dhomse et al. (2011). Briefly, the MLR has 52 terms, including 12 monthly linear trend terms, 24 QBO terms (at 30 and 50 hPa) as well 12 age-of-air (AoA) TOMCAT tracer terms to account for inter-annual dynamical variability. For solar flux variability, we include the composite Mg-II index from University of Bremen, Germany, via http://www.iup.uni-bremen.de/UVSAT/Datasets/mgii (Snow et al., 2014). El Nino/ Southern Oscillation (ENSO), Arctic Oscillation (AO) and Antarctic Oscillation (AAO) index terms are also included to account for effects of important tele-connection patterns. QBO, ENSO, AO, and AAO indices are obtained from Climate Prediction Center, via https://www.cpc.ncep.noaa.gov/ (last access: 15 May 2021). To simplify interpretation of regression coefficients, excluding 12 linear trend terms, all the explanatory variables are detrended and normalised between 0 and 1. As F10.7 solar flux changes over the 2005–2020 time period are about 99.4 units, estimated SCS using normalised Mg-II index can be considered to be the same as SCS per 100 solar flux units.

MLR models include various explanatory variables to separate the influence of individual processes, but they are required to be completely independent. However, to some extent most atmospheric processes are coupled. Hence, most previous studies have used OLS regression models that suffer from multi-collinearity issues. For example, the two QBO terms used here as well as in various earlier studies are not completely independent. Dynamical proxies such as age-of-air (or eddy heat fluxes in Dhomse et al. (2006)), are also coupled with the QBO phase via the Holton-Tan mechanism (Holton and Tan, 1982). Additionally, OLS models are designed to minimise errors but have relatively high variance. This means even slight changes in explanatory variables lead to large changes in the estimated regression coefficients. Therefore, we use an ensemble of regularised least squares (RLS) models. RLS models constrain or shrink regression coefficients to reduce the variance. Ridge regression (or L1 regularisation) uses Tikhonov regularisation (Hoerl and Kennard, 1970), where coefficients for all the parameters are scaled down with optimum weight or penalty term. In contrast, Lasso regression (L2 regularisation, Tibshirani, 1996)

uses the square of the penalty term to scale down the regression coefficients. ElasticNet regression (Zou and Hastie, 2005) combines the strengths of Lasso and Ridge regression to scale down the regression coefficients. Regression models used here are from Python scikit module (Pedregosa et al., 2011). For details see https://scikit-learn.org/stable/modules/linear_model.html (last access: 30 July 2021)

## 4 Results

Different combinations of multivariate regression models are used to estimate long-term ozone trends as well as to quantify the influence of important processes on ozone variability (e.g. Braesicke et al., 2018; Petropavlovskikh et al., 2019). Here, we use identical regression models to estimate the SCS in stratospheric ozone from MLS and the model simulations described above. Figure 1 compares MLS ozone anomalies and OLS MLR-fitted regression lines near the equator (1.5° latitude) at 9 pressure levels. As expected, the largest ozone variability ($\approx \pm 15\%$) is observed in the lower stratosphere (46.4 hPa) and its magnitude declines almost linearly to higher altitudes except 14.6 hPa. Minimum variability seen near 14.6 hPa is somewhat puzzling and one possible explanation might be the damping effects of the QBO and semi-annual oscillation-related ozone variability near these levels. Overall, the regression lines show excellent agreement with monthly MLS ozone anomalies ($R^2 > 0.5$) and the residuals are less than a few percent at all levels. Somewhat larger residuals (up to 5%) occur near 46 hPa (though $R^2 \geq 0.85$), indicating that even with 24 QBO terms, the regression model has some difficulty in capturing some of the QBO-related ozone variability due to the unusual QBO behaviour over the last decade (e.g. Osprey et al., 2016; Anstey et al., 2021).

Figure 2 shows the MLS observation-based SCS (2005–2020) for the tropical latitude band (20°S–20°N). The SCS estimated using HALOE (1992–2005, volume mixing ratio, vmr), SAGE II (1984–2005, vmr), SAGE II (1984–2005, number density) and SBUV (1979–2005, vmr) presented in Dhomse et al. (2011, 2015) are also shown for direct comparison. Figure 2 clearly shows that the MLS-based SCS is significantly different to that from all other datasets, although with some similarity to the HALOE-based SCS. A key feature is that the MLS SCS shows a clear broad positive peak in the mid-upper stratosphere that is almost twice as large as any other satellite-data-based SCS reported in the past (e.g. Soukharev and Hood, 2006; Remsberg and Lingenfelser, 2010). On the other hand, our estimates are somewhat consistent with the latest BASIC v2-based estimates (1984-2016) presented in Ball et al. (2019), though MLS shows a ∼50% larger peak around 40 km against around 35 km in the BASIC data. However, it is important to note that for the 2004–2016 time period, MLS data is used in the BASIC v2 reconstruction. Hence differences between our SCS estimates and that presented in Ball et al. (2019) could be due to using a longer time series (extended time period) or the aliasing effects of other processes (volcanoes, EESC changes).

Near the stratopause region (around 50 km), only MLS and HALOE show a SCS of less than 1%. The clear difference between MLS versus SAGE II, HALOE and SBUV could be due to a combination of various factors. First, as SAGE and HALOE use the solar occultation technique, even under ideal conditions they provide only about 900 profiles per month over the whole globe. Hence, fewer and sparser profiles are used to calculate monthly mean profiles. In contrast, MLS is a thermal emission limb sounder with a few hundred thousand profiles available for monthly mean calculations. Hence, the MLS-derived SCS suffers minimal impact from non-uniform temporal sampling compared to SAGE and HALOE (e.g. Toohey et al., 2013;

Sofieva et al., 2014; Millán et al., 2016). Second, the HALOE and SAGE II data cover a period that has non-linear changes in the equivalent effective stratospheric chlorine loading (EESC, e.g. Newman et al., 2007; Engel et al., 2018a), whereas MLS covers a period where EESC is decreasing almost linearly in response to the actions taken under the Montreal Protocol (e.g. Kohlhepp et al., 2012; Strahan and Douglass, 2018). Third, all of the satellite ozone retrieval algorithms rely on meteorological (re)analysis datasets for the background atmospheric state. Therefore, with technological advances as well as the huge increase in the number of assimilated meteorological observations, the MLS retrieval scheme might have some advantage over the earlier data records. Fourth, the eruption of Mt Pinatubo in June 1991 injected about 14-23 Tg $SO_2$ into the stratosphere (e.g. Guo et al., 2004), leading to significant enhancement in the stratospheric aerosol layer for few years. The enhanced stratospheric aerosols lead to larger ozone retrieval errors for occultation instruments, particularly in the lower stratosphere (e.g. Wang et al., 1996; Thomason, 2012). Enhanced stratospheric aerosols from Mt Pinatubo also caused significant ozone losses and changes in the stratospheric circulation (e.g. Dhomse et al., 2015, 2020) that could have had an impact on the SCS estimates. Fifth, MLS observations cover the recent solar cycle (number 24, 2009–2020), which is one of the weakest cycles (fewer sunspots) over the last century, hence SSI changes may have been somewhat different than for earlier solar cycles. However, a weaker solar cycle does not mean that the SCS during previous cycles would be larger as complications also arise from various complex couplings such as temperature feedback (increased direct radiative heating during solar maxima), wavelength-dependent photolysis rates (irradiance changes are not uniform across different wavelengths). Sixth, SBUV and SBUV/2 are nadir-viewing instruments with very coarse vertical resolution, especially in the upper stratosphere, which can lead to different (and smoother) SCS profiles.

Another very important difference is observed in the lower stratosphere, where MLS suggests a much smaller ($\pm 1\%$) SCS compared to about 5% SCS in the SAGE II data. It has long been postulated that the lower stratospheric SCS is most probably due to the aliasing effect of volcanic eruptions, QBO and ENSO (e.g. Lee and Smith, 2003; Chiodo et al., 2014). In fact, Dhomse et al. (2011) clearly showed that a CTM simulation with annually repeating dynamics produced a secondary peak in the tropical lower stratosphere that was significantly smaller when simulations are performed with fixed stratospheric aerosols. As there have been no significant volcanic eruptions during the MLS period, this suggests that the large positive SCS in the tropical lower stratosphere reported in SBUV and SAGE II-based studies is likely due to non-linear changes in EESC and influences from strongly perturbed stratospheric aerosol layer leading abrupt changes in ozone chemistry and stratospheric dynamics following major volcanic eruptions. Later, we show that a simulation with fixed dynamics does not show a secondary peak for the 2005–2020 time period.

We performed the MLS-like analysis on TOMCAT-simulated ozone profiles from runs **A_NRL**, **B_SAT**, **C_SOR** and **D_SFix** for all 64 latitude bands and 36 pressure levels ranging from 300 to 0.1 hPa. Comparisons between model (including **E_DFix**) and MLS tropical (20°S–20°N) ozone anomalies at five different pressure levels are shown in Figure 3. Overall, anomalies from the first three simulations (**A_NRL**, **B_SAT** and **C_SOR**) show very similar ozone variations, and mean ozone differences in the tropics are within $\pm 1\%$ at all pressure levels. An important aspect in Figure 3 is that even at 1 hPa modelled ozone differences are always less than 1%, suggesting consistency between all three solar flux datasets. This clearly highlights

that earlier studies showing large negative SCS simulated using SORCE data (e.g. Haigh et al., 2010; Merkel et al., 2011) must have predicted unrealistic ozone variations due to biases in SORCE data as well as much shorter MLS time series.

Additionally, anomalies from **D_SFix** and **E_DFix** illustrate the effects of solar flux variations. For example, in the lower stratosphere **E_DFix** shows much smaller variations while **D_SFix** anomalies are very similar to **A_NRL** anomalies, confirming exclusive dynamical influence on the ozone variability. However, in the mid-upper stratosphere **D_SFix** anomalies are clearly smaller than **A_NRL**, and **E_DFix** anomalies show variations of about $\pm 2\%$. In order to better understand the effects of time-varying solar fluxes we performed composite and correlation analyses on detrended tropical (20°S-20°N) ozone anomalies. Figure 4 shows ozone composites for solar maximum and minimum months, as well as the correlation between tropical ozone anomalies and the Mg-ii index. Solar maximum/minimum months are calculated by selecting months when the Mg-ii index is higher/lower than one standard deviation. The composite and correlation analyses clearly indicate that **A_NRL** and MLS-derived estimates are in excellent agreement. As expected, **A_NRL** shows up to 3% ozone increase during solar maximum that is almost exclusively because of solar flux variations (**E_DFix**). An important feature in Figure 4 is that **D_SFix** ozone anomalies show very little change between solar maximum and solar minimum months suggesting the implicit SCS in ERA5 dynamics is not enough to simulate observed (MLS-based) ozone variations in the middle and upper stratosphere. As expected run **E_DFix** anomalies show very high correlation with Mg-ii index throughout the stratosphere. In contrast, both MLS and **A_NRL** correlation are close to each other with peak values of about 0.3 near 4 hPa and **D_SFix** anomalies show very little or negligible correlation with the Mg-ii index. Again, this confirms that ERA5 dynamical fields contain only little or no implicit SCS.

Figure 5 shows SCS estimates for four model simulations as well as MLS data using OLS and three regularised (Lasso, Ridge, and ElasticNet) regression models. As expected, regression coefficients from the three regularised models are somewhat smaller than OLS estimates, but overall all the regression models show consistent behaviour. Some key features are a maximum SCS near the tropical and mid-latitude mid-upper stratosphere (near 4 hPa or 40 km) and a negative SCS in the low- and mid-latitude lower stratosphere. It is important to note that the MLS and model-based (**A_NRL**, **B_SAT** and **C_SOR**) SCS are larger than 1-$\sigma$ uncertainty in the tropical and mid-latitude middle stratospheric region (between 30 and 3 hPa). Larger uncertainty in the estimated SCS at the high latitude lower stratosphere must be due to the relatively short available time series (16 years) and large interannual variability in those regions. Additionally, a second lobe of positive SCS extending from the tropical middle stratosphere to the Arctic lower stratosphere (near 50 hPa) is clearly visible in all the panels. This is consistent with an earlier analysis by Labitzke and Loon (1988).

However, an unexpected feature is that except for the Ridge regression model, a large SCS near the Antarctic lower stratosphere is visible in all the models. To our knowledge, this type of strong SCS in the Antarctic stratosphere has not been reported in earlier studies. It could be due to a combination of various factors. First, most of the earlier studies used SBUV, SAGE or HALOE datasets that have limited coverage during dark polar night. Second, the sudden stratospheric warming in the 2019 Antarctic polar vortex stratosphere (e.g. Lim et al., 2020) and wave activity in other recent years, as well as ongoing EESC decreases, might have caused the aliasing effect for the SCS estimation. Actually, close inspection of **D_SFix**-based estimates suggests that half of the SCS in the Antarctic lower stratosphere may be of dynamical origin.

Another important feature in Figure 5 is that in the lower stratosphere, all four model simulations (and MLS data) show large negative SCS confirming significant dynamical influence and very little effect from time-varying solar fluxes. However, we find that model-simulated and MLS SCS vary significantly in the mid-latitude lower stratosphere. In the SH mid-latitudes, MLS data suggest up to -6% SCS whereas the model simulations suggest only up -2%. On the other hand, in the NH mid-latitudes, MLS data suggest negligible SCS, but model simulations show -4% SCS. These inter-hemispheric differences between model and MLS data might be due to discrepancies in the ERA5 reanalysis data set that is used to force TOMCAT (e.g. Chrysanthou et al., 2021).

Figure 6 shows a comparison between the mean SCS for the tropics (20°S-20°N) from the four different regression models shown in Figure 4. As seen in Figure 1, MLS shows the largest SCS near 4 hPa and all the model simulations also show similar SCS profiles. Note that the SCS based on runs **A_NRL** and **B_SAT** show nearly identical behaviour. This suggests that although there are non-negligible differences between the construction of the NRL2 and SATIRE solar irradiances (e.g. Yeo et al., 2014; Matthes et al., 2017; Coddington et al., 2019), their wavelength-dependent differences seem to cancel out to produce a nearly identical SCS in stratospheric ozone. In terms of magnitude, OLS-based estimates suggest that MLS shows up to a 3% SCS near 4 hPa (~40 km), while the NRL2 and SATIRE peaks are about 4.5% and the SORCE peak lies between the MLS and NRL2 estimates. An important feature in Figure 5 is that even with regularisation, the MLS-based SCS does not show a significant reduction or alternation, confirming the robustness of the estimated SCS. Most importantly, in Figure 4 run **D_SFix** suggested an almost negligible chemical SCS near 4 hPa, but the regression model suggests a SCS of up to 1% at this altitude demonstrates possibility of some implicit SCS ERA5 dynamical fields.

In the lower stratosphere (below 25 km) all the simulations show a smaller (or more negative) SCS compared to MLS. As expected, the regularisation models (Lasso, Ridge and ElasticNet) do not change the profile structure significantly but the estimated magnitudes are somewhat smaller in magnitude with similar behaviour in the 3 model simulations. Interestingly, even with regularisation the MLS-based SCS does not turn negative in the upper stratosphere, indicating that earlier SORCE-based studies (e.g. Haigh et al., 2010; Merkel et al., 2011; Ball et al., 2016) were likely impaired by their shorter timescales as well as biases in an earlier version of the SORCE dataset.

Finally, as run **C_SOR** also shows good agreement with MLS-based SCS (though within associated uncertainties) similar to runs **A_NRL** and **B_SAT**, we analyse the difference between these model simulations. Figure 7 shows tropical (20°S-20°N) percentage ozone differences between three model simulations with time-varying solar fluxes (**A_NRL**, **B_SAT** and **C_SOR**) and the simulation with fixed solar fluxes (**D_SFix**, which uses the mean 2005-2020 NRL V2 fluxes). As expected, all comparisons show the largest ozone difference in the mid-upper stratosphere. The time-varying solar flux simulations show a steady decline in ozone differences until 2008 and positive ozone changes after 2011 (solar maximum), followed by an ozone decrease after 2016. Interestingly, run **C_SOR** show much larger positive differences during 2004/2005 that hardly turn negative in 2008, but show up to -3% ozone differences in 2016. As seen in Figure 6, both runs **A_NRL** and **B_SAT** show a similar pattern in ozone differences, though the magnitude of ozone change is somewhat larger in run **B_SAT**. A somewhat different structure in ozone difference during maxima and minima might be due to differences in absolute solar fluxes.

The most interesting aspect in Figure 7 is that near 5 hPa, run **C_SOR** shows up to +3% ozone difference between 2005–
2008 compared to about +2% in runs **A_NRL** and **B_SAT**. Similarly, after 2016 run **C_SOR** shows ozone differences over -3%
in magnitude in the mid-upper stratospheric ozone which is around 1.5× larger than runs **A_NRL** and **B_SAT**. So, although
there are significant variations in the ozone difference ~~patterns~~magnitudes, various model simulations clearly show that all of
the solar flux datasets lead to similar patterns in ozone variation, i.e. ozone increases towards solar maxima followed by steady
decline towards solar minima. Thus, the results from composite analysis are consistent with the regression analysis. However,
the magnitude of ozone variations with respect to the NRL V2-based fixed solar flux simulation is almost double in a simulation
with SORCE solar fluxes, whereas regression analysis suggests run **C_SOR** has a weaker ~~SC~~ SCS in the low-mid stratosphere.
This clearly highlights that model-simulated ozone changes depend on both magnitude of solar irradiances as well as their time
variations. Most importantly, the somewhat different (and non-linear) ozone differences seen in **C_SOR** suggests that SORCE
solar fluxes may still have some time-varying biases. The larger UV variability reported in earlier versions of the SORCE data
(see Section 1) is reduced but apparently still larger than that given by SATIRE or NRL v2.

## 5 Conclusions

Our key result is that we have presented an analysis of the solar cycle signal (SCS) in stratospheric ozone based on MLS
v5 satellite data (2005-2020). Previously, our understanding of the ozone SCS has been largely based on 22 years of SAGE
II v7 data (Dhomse et al., 2016; Maycock et al., 2016). As the MLS satellite instrument has a much better spatial coverage
than any other ozone dataset providing more than 16 years of continuous ozone profile measurements, it is ideally suited for
re-evaluating our understanding of the processes controlling/modifying stratospheric ozone. MLS data also covers a period
where EESC changes are almost linear and there has been no major volcanically induced perturbation to the stratospheric
aerosol layer. Hence the SCS attribution is relatively cleaner than in previous datasets where trends as well as attribution are
complicated as they include periods with strong volcanic eruptions.

Our analysis suggests a single-peak-structured SCS in the tropical stratosphere, which is significantly different to that derived
in previous studies based on SAGE II and SBUV datasets during earlier periods. In contrast, the MLS-based SCS shows a
similar structure to that from HALOE data, although its peak amplitude near 3 hPa is almost double that of HALOE (up to
3%). The lack of a secondary peak in MLS satellite data suggests that the Mt. Pinatubo volcanic eruption induced chemical
and dynamical changes which caused an aliasing effect in the estimated SCS. This analysis is consistent with the postulations
discussed in modelling studies such as Lee and Smith (2003), Dhomse et al. (2011) and Chiodo et al. (2014).

We also performed three model sensitivity simulations with different solar flux datasets: NRL2, SATIRE and SORCE. We
find that the SCS from the simulation with SORCE fluxes is somewhat smaller in magnitude but is within the uncertainties
seen in the MLS-derived SCS as well as NRL2 and SATIRE data. Overall, all three model simulations show SCS structures
very similar to that in MLS data. Importantly, it suggests that with recent adjustments and corrections (Harder et al., 2019),
SORCE data can be used to study the effects of solar flux variations, though some time-varying biases in SORCE data cannot
be ruled out. We also performed an ensemble of linear regression models (OLS, Lasso, Ridge and ElasticNet) that confirm

the robustness of the SCS. All of the regression models show a broad peak near low-mid latitudes around 4 hPa. MLS data and model simulations also indicate a much larger SCS in the Antarctic stratosphere that could be due to the aliasing effect of ozone recovery due to a decrease in EESC loading as well as changes in stratospheric transport in the SH. Finally, regression and composite analyses of model simulations with respect to fixed solar flux simulations suggest that both absolute magnitude as well as time variations in solar flux forcing data sets play key roles in SCS estimates.

*Data availability.*  MLS data is publicly available via https://disc.gsfc.nasa.gov/. TOMCAT data can be downloaded from http://homepages. see.leeds.ac.uk/~fbsssdh/TOMCAT_SOLAR/. NRLV2, SATIRE and SORCE Solar Irradiance data Center https://lasp.colorado.edu/lisird/. Solar activity proxy index (Mg II index) is available at http://www.iup.uni-bremen.de/UVSAT/Datasets/mgii. QBO, ENSO, AO, AAO indices are obtained from Climate Prediction Center, via https://www.cpc.ncep.noaa.gov/.

*Competing interests.*  Authors have no competing interests.

*Acknowledgements.*  We are grateful to William Ball for useful comments. SSD and MPC were supported by the NERC SISLAC project (NE/R001782/1) and NCEO (NE/R016518/1). We thank NASA for MLS v5 data. Work at the Jet Propulsion Laboratory, California Institute of Technology, was carried out under a contract with the National Aeronautics and Space Administration. We thank the European Centre for Medium-Range Weather Forecasts for providing their analyses. The model simulations were performed on the UK national Archer and Leeds Arc4 HPC systems.

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

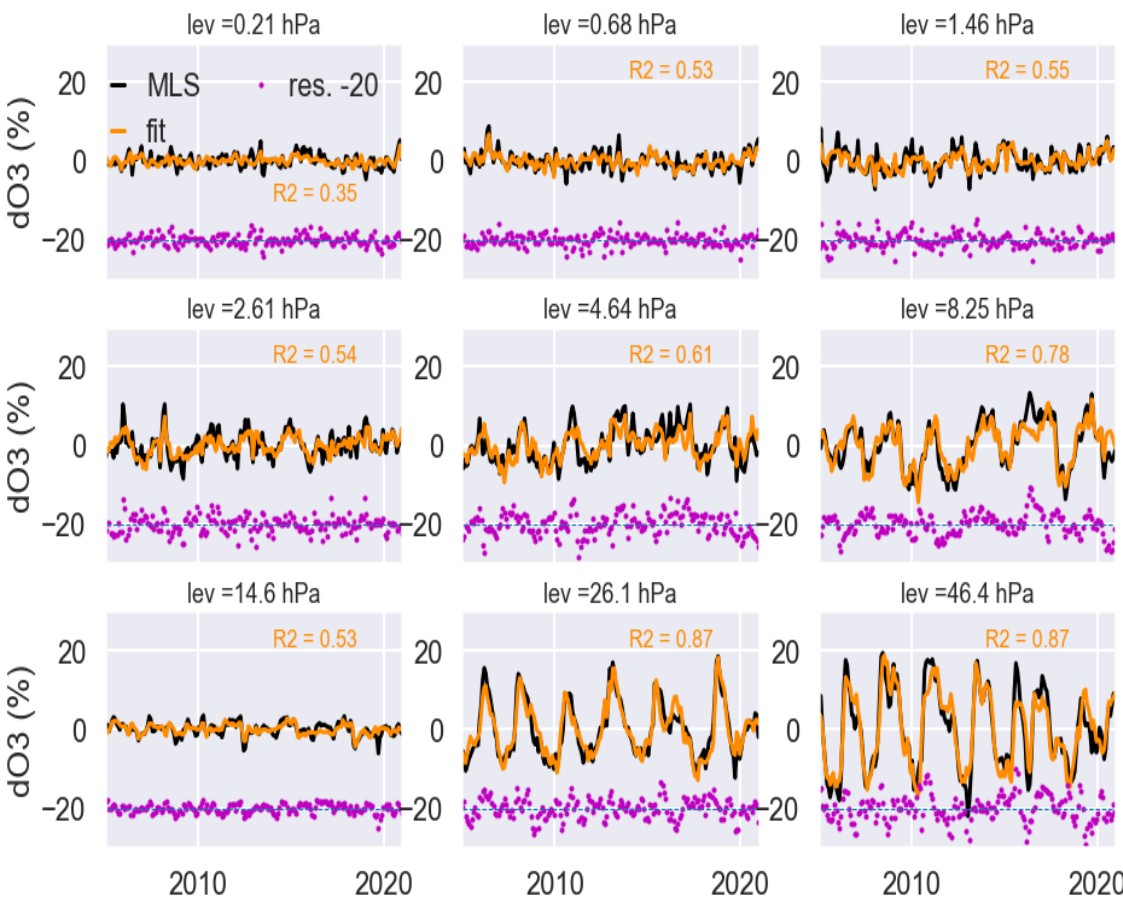

**Figure 1.** Monthly mean ozone anomalies from MLS V5 (black line) for 2005-2020 and corresponding regression fits (orange line) for nine different pressure levels at 1.5°N. Goodness of fit ($R^2$) values are also shown with dark-orange-coloured text and residuals are shown at the bottom of each panel as pink dots. For clarity the residuals are shifted by -20%.

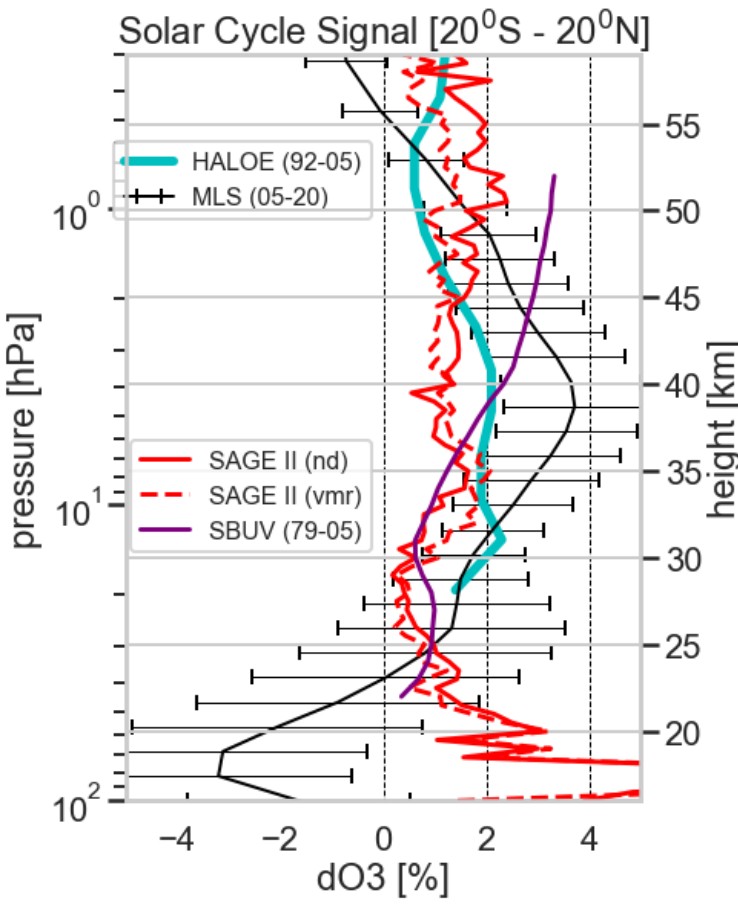

**Figure 2.** Comparison of ozone solar cycle signal (SCS) from various satellite data products for the tropical (20°N–20°S) region. SCS derived using SAGE II V7.0 [1984-2005] data in terms of number density and mixing ratio units (Dhomse et al., 2016) are shown with solid and dashed red lines, respectively. SCS from HALOE (1992–2005) and SAGE-corrected SBUV (McLinden et al., 2009) (1979–2005) datasets are shown with aqua and purple lines, respectively (Dhomse et al., 2011). SCS from MLS V5 data (2005–2020) is shown with the black line.

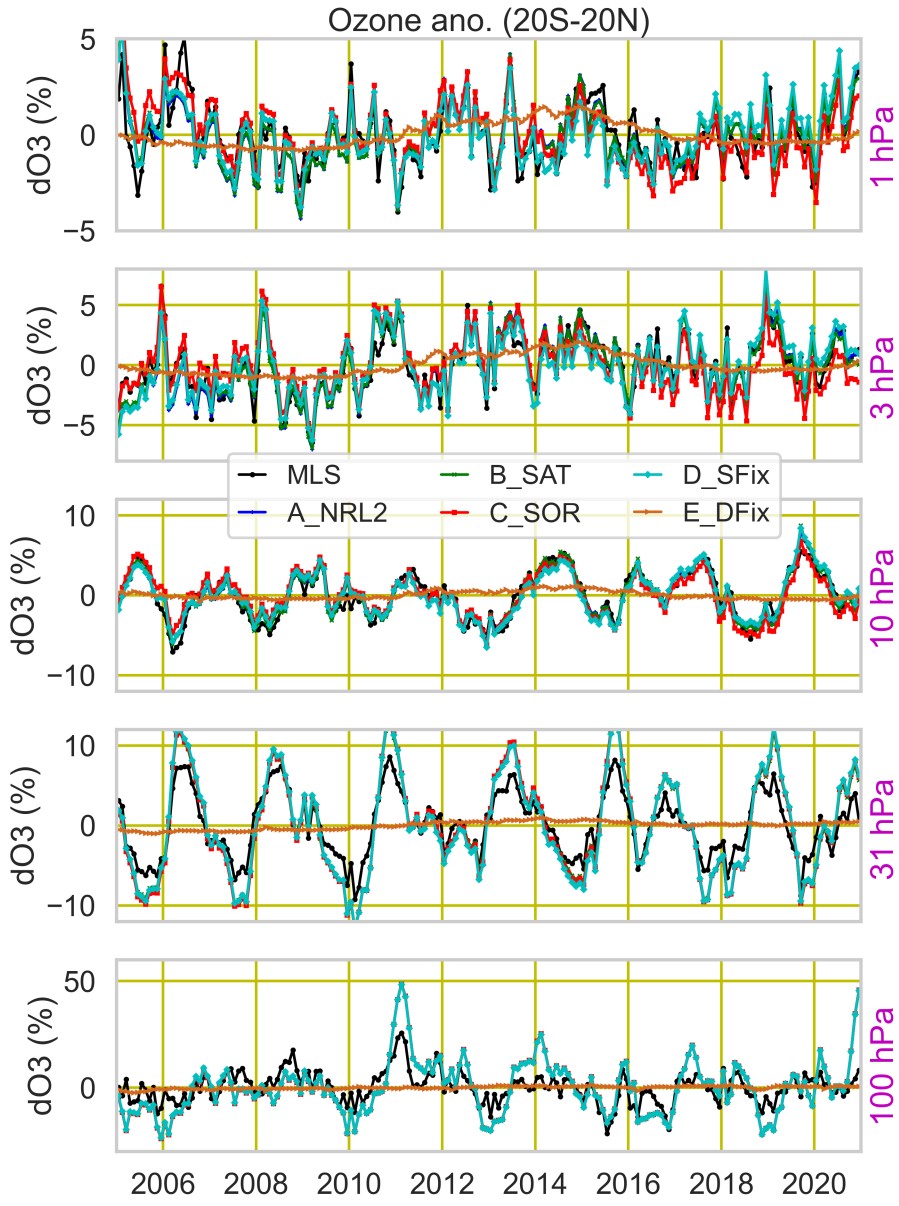

**Figure 3.** Monthly mean ozone anomalies (%) from MLS V5 (black line) and five TOMCAT model simulations for the tropics (20°S-20°S) for 2005-2020. Ozone anomalies from simulations with NRL V2 (Coddington et al., 2016), SATIRE (Yeo et al., 2014) and SORCE (Harder et al., 2019) are shown with blue, green and red lines, respectively, whereas anomalies from fixed solar fluxes and fixed dynamics (year 2004) are shown with cyan and orange colours. Anomalies are shown for five pressure levels (top to bottom): 1 hPa, 3.1 hPa, 10 hPa, 31 hPa and 100 hPa.

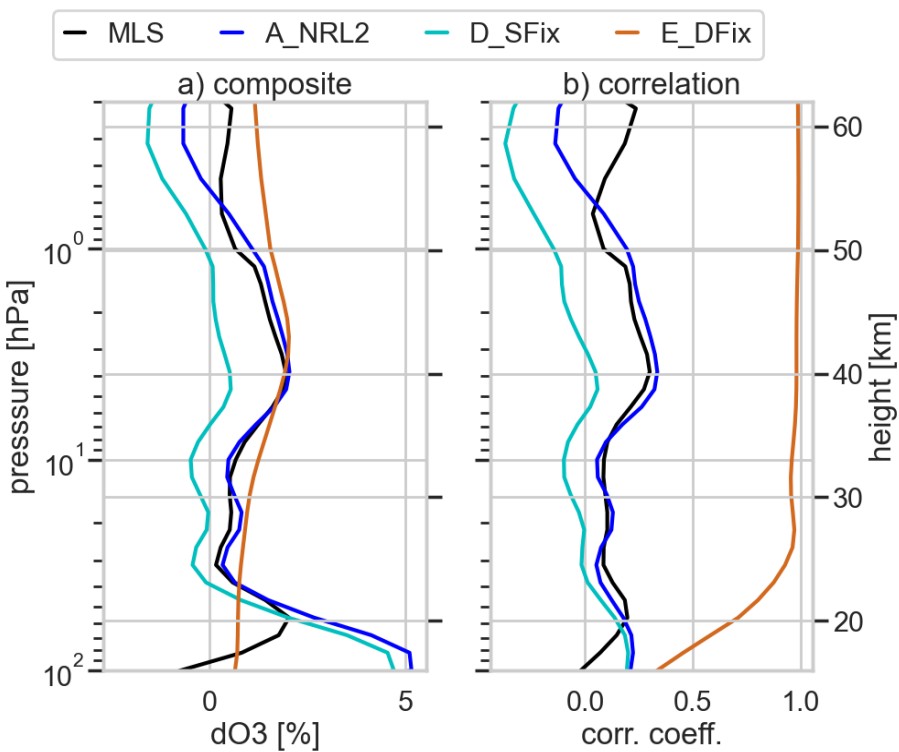

**Figure 4.** (a) MLS ozone profile composite for solar maxima (n=40) and solar minima (n=51) months (black line). Percentage ozone differences for the tropics (20°N–20°S) between a model simulation with time-varying NRL2 solar flux (**A_NRL**), fixed solar flux (**D_SFix**) and fixed dynamics (**E_DFix**) simulations are shown with blue, cyan and orange lines, respectively. (b) Correlation coefficient between Mg-ii index and monthly mean ozone anomalies from MLS and simulations **A_NRL**, **D_SFix** and **E_DFix**.

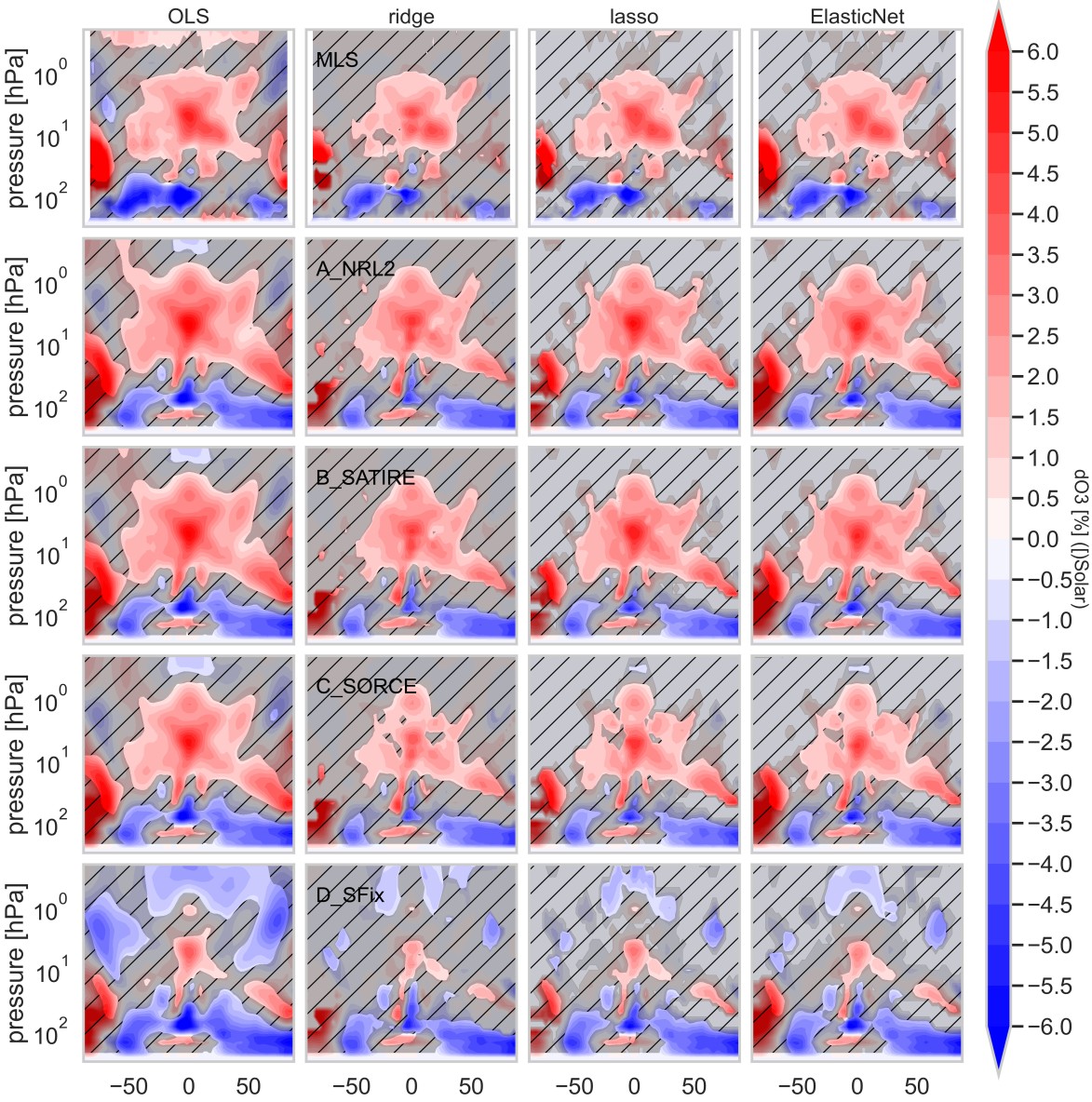

**Figure 5.** Latitude-pressure cross sections of solar regression coefficients (or Solar Cycle Signal per 100 solar flux units) for MLS (top row) as well as model simulations **A_NRL** (second row), **B_SAT** (third row), **C_SOR** (fourth row) and **D_SFix** (bottom row). Regression coefficients are from OLS (first column), Lasso (second column), Ridge (third column) and Elastic Net (fourth column) regression models. Stippling indicates regions where regression coefficients are smaller than 1-$\sigma$ uncertainty estimates.

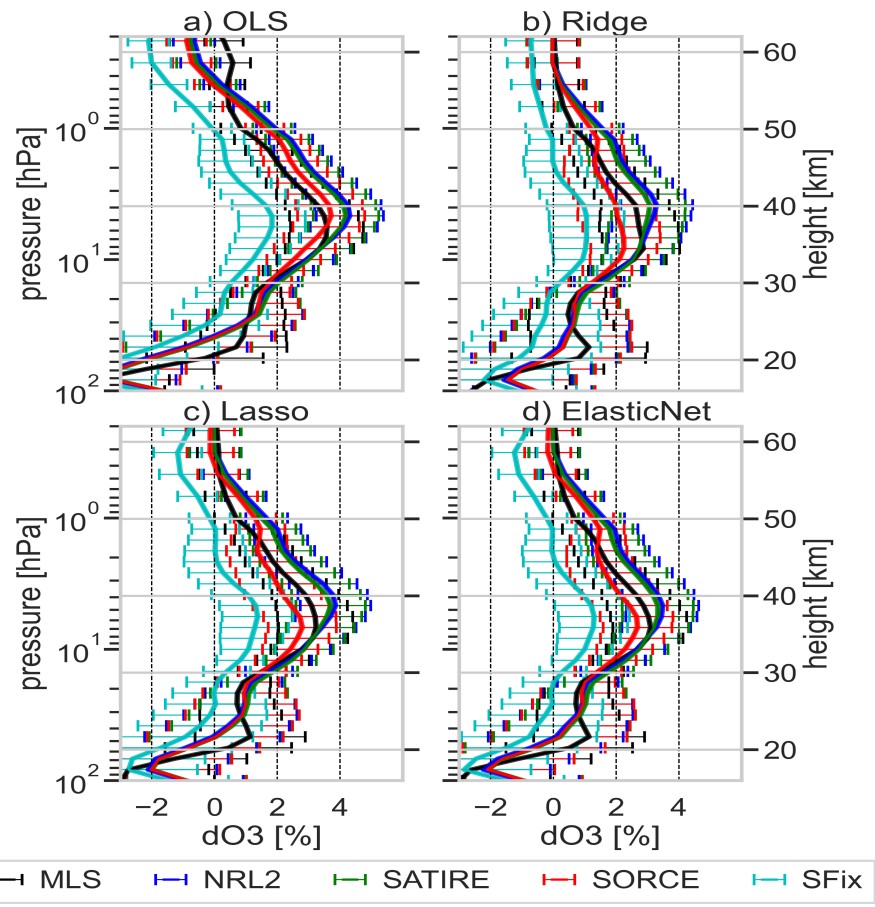

**Figure 6.** Solar cycle signal (SCS) for 2005-2020 period (per 100 solar flux units) in tropical (20°N–20°S) stratospheric ozone from MLS and ozone profiles from four model simulations (**A_NRL**, **B_SAT**, **C_SOR** and **D_SFix** ) using four types of regression models (a) OLS, (b) Lasso, (c) Ridge and (d) Elastic Net. Horizontal lines show averaged 1-$\sigma$ uncertainties.

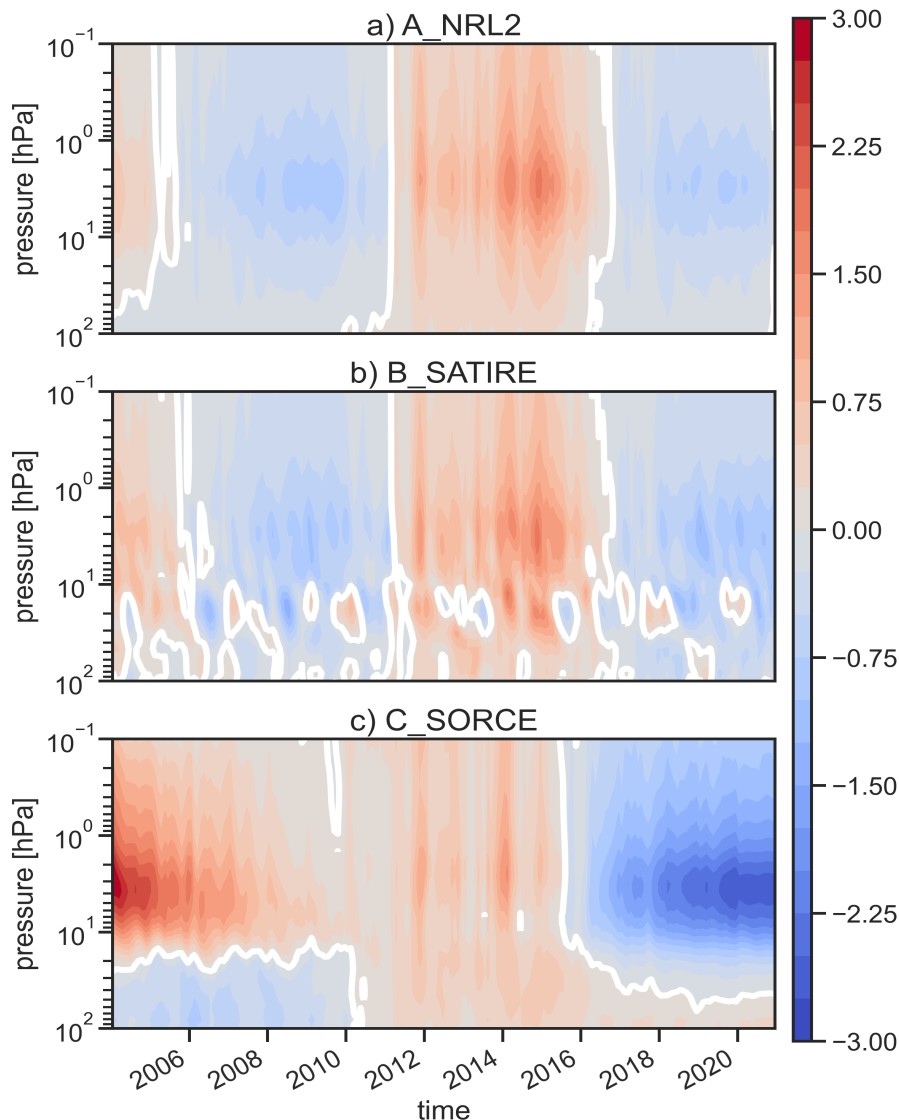

**Figure 7.** Percentage difference in tropical ozone (20°N–20°S) between a model simulation with time-varying solar flux and a simulation with fixed solar flux for (a) NRL2, (b) SATIRE and (c) SORCE. White-coloured lines show zero contours.