# Peer review of "A Single-Peak-Structured Solar Cycle Signal in Stratospheric Ozone based on Microwave Limb Sounder Observations and Model Simulations"

_Atmospheric Chemistry and Physics, 2021_

## Author Comment (AC1)

**Replies to Reviewer 1**:

General Comments:

The authors use Microwave Limb Sounder (MLS) instrument measurements to estimate the 11-year solar cycle signal (SCS) in stratospheric ozone. Their analysis of the MLS data suggests a single-peak-structured SCS signal of about 3% near 4 hPa (~40 km) in tropical stratospheric ozone. This finding is significantly different from earlier work that found a double-peak-structured SCS, which was based on ozone profiles from Stratospheric Aerosol and Gas Experiment (SAGE) II or Solar Backscatter Ultraviolet Radiometer (SBUV) satellite instruments' data. They also found that MLS-observed ozone variations are more consistent with ozone from a control model simulation using Naval Research Laboratory (NRL) v2 solar fluxes. The lowermost stratosphere modelled ozone shows a negligible SCS, which is somewhat different from the nearly 1% variation derived using MLS data.

This article contains a good thorough description of previous work on the SCS in stratospheric ozone. The presented work is then given in context with the published literature and shows a good analysis and comparison of the SCS in measurements and model simulations. It is significant that the research includes model sensitivity simulations with three different solar flux datasets (NRL2, SATIRE and SORCE). Also, it is noteworthy that an ensemble of four linear regression models were used to test the derived robustness of the SCS.

I do think that the paper should be published.

**We thank the reviewer for his/her comments.**

Specific Comments:

1) p. 9, lines 257-258: The sentence 'Most importantly, somewhat different (and non-linear) ozone differences seen in C_SOR suggests that SORCE solar fluxes may still have some time-varying biases' is quite important. Does this mean that the SORCE solar fluxes still possibly overestimate UV variability?

**We are not sure if they overestimate UV variability. However, UV variability between NRL, SATIRE and SORCE solar fluxes is compared in Harder et al., (2019) (Figure 12). Their comparison shows that SORCE data does show larger UV variability during both the solar cycles especially during recent solar cycle (24); SORCE data suggest large changes for SSI between 300 and 380 nm.**

We also added a sentence to explain these biases, "The larger UV variability reported in earlier versions of the SORCE data (see Section 1) is reduced but apparently still larger than that given by SATIRE or NRL v2"

2) p. 20, Figure 5: My eyesight is not the best and I have a minor problem distinguishing the two different colors used to present the results from MLS observations (black) and a model simulation with NRL2 solar fluxes (dark blue). The 'black' and 'dark blue' look very similar in color to me. Would it be possible to use a 'lighter blue' color for the model simulation or even a 'dashed black' line for the MLS observations? This would aid those of us with poorer vision.

**We agree with the reviewer. In the revised manuscript A_NRL lines are shown with light blue colour.**

*References:*

*Harder JW, Béland S, Snow M. SORCE-based solar spectral irradiance (SSI) record for input into chemistry-climate studies. Earth and Space Science. 2019, 2487-2507.*

---

## Author Comment (AC2)

**Replies to Reviewer 2:**

In the paper "A single-peak-structured solar cycle signal based on Microwave Limb Sounder observations and model simulations" by Dhomse et al., the authors employ a multivariate regression on 16 years of ozone observations from MLS to derive the solar cycle signal. The same analysis is performed on model experiments with a chemistry-transport model driven by ECMWF re-analysis data, using different data-sets of solar spectral variability. They find a clear solar cycle signal with a single-peak structure and significantly higher amplitude than in previous estimates based on other data-sets, and they argue that this is due to a combination of higher sampling rate of the MLS data compared to, e.g., solar occultation instruments used in previous studies, and to less aliasing with other modes of atmospheric variability, in particular stratospheric halogens and volcanic forcing. They also find really excellent agreement between the solar cycle signals derived from observations and all model results shown, suggesting that all forcing data-sets used can be used in studies of solar-climate interactions. This is a careful analysis of an interesting and promising data set concerning a very interesting (if difficult) question, and the paper is generally clearly structured and well written.

**We thank the reviewer for his/her encouraging comments. Our replies to the comments are in red coloured text below.**

However, there are two points that I think should be addressed before final publication. First of all, while the point raised about less aliasing with volcanic forcing and stratospheric halogen loading appears plausible, the fact that 24 fitting factors were needed for the QBO signal suggests to me that an aliasing with the QBO is possible, and this should be investigated / discussed in more detail.

Second, both MLS observations and model results show consistent solar signals in the lower stratosphere with a distinctive latitudinal structure. This appears unlikely to be due to direct solar forcing, and more likely due to a dynamical feedback which would be implicitly included in the model results due to the use of dynamical fields from reanalysis data. This could be investigated simply by doing the same multivariate analysis on a model run with constant solar forcing which already exists, and I would urge the authors to do this. These points are discussed in more detail below, as well as a few more minor points.

Lines 100 – 102: I have been wondering here about the justification of using a chemistry-transport model driven by re-analysis data. Superficially, this could be understood to mean that the solar cycle signal derived from the model experiments is purely the chemical response of the atmosphere. However, any potential dynamical feedback in the atmosphere is implicitly contained by the use of the re-analysis data, and this will obviously also affect the ozone fields, by transport and by its dependence on temperature. But the reverse feedback, from the ozone fields to dynamics via radiative heating is suppressed to some extend by the use of prescribed temperatures and wind fields. Does that mean that the ozone fields and model dynamics are not fully consistent with each other? I'm not quite sure, but would have liked a discussion of this somewhere. On the other hand, using the same dynamical situations with and without variable solar forcing provides the interesting possibility to separate chemical responses from dynamical feedbacks. I'll come back to this later on.

Lines 155-162: Looking at Figure 1, it seems obvious that the QBO provides the largest source of ozone variance in most pressure layers, probably much larger than the comparatively small solar cycle signal. Considering that you fit 24 different QBO terms, and that the residual of the fit appears to be several percent, in the same order of magnitude, or even larger than, the solar cycle signal of 1-4 % that you derive (see Figure 2) – how confident can you be that the solar cycle signal is not affected by aliasing with the many QBO terms, or their superposition? Please add some analysis/discussion on this point. You

could, e.g., show a comparison for the amplitudes of the different terms (QBO, linear term, SCS, volcanic), for the pressure levels given in Fig. 1, compared to the residuals.

Line 201-207, discussion of differences between regression analyses of the different model experiments, Figure 3: again, the largest mode of variability appears to be the QBO signal. I would expect that this is very similar in all model experiments, and that results of D_SFix would be very consistent with the other model experiments here. Why not include those in the figure? On the other hand, there are quite significant differences between model results and observations in 31 hPa and 100 hPa. Can you discuss where those derive from? As model results A_NRL, B_SAT and C_SOR are nearly identical in these pressure levels, so presumably not due to any chemical solar cycle signal. To highlight the impact of the solar cycle, you could show the differences A_NLR-D_SFix, B_SAT-D_SFix, C_SOR-D_SFix on these pressure levels as well.

Lines 208 – 222, discussion of Figure 4: I had to look very carefully at Figure 4 to ensure that the results from the different data sets, and particularly MLS and NRL2, are not identical. The agreement of the patterns of positive/negative SCS between observations and model results is really striking. However, you mention in the text that results are not statistically significant everywhere. Could you a) describe how statistical significance was derived (e.g., by explaining the errors shown further up), and b) somehow mark regions of significance in the figures? Also, you show in Figure 3 that the model results at 31 hPa and 100 hPa are virtually identical, but the regression results shown here are not – why is this the case? Does the fact that the model results are nearly identical mean that there is no chemical solar cycle signal, or just that the chemical solar cycle signal is identical in the lower stratosphere? This is not possible to see from the results shown, but could be obtained by comparing to results of the model run D_SFix. So again – could you add results of D_SFix and differences, to Figure 3 as suggested above? Anyway I would expect that any solar cycle signal in the lower stratosphere is not a direct response of the chemistry, but due to dynamical feedbacks which the model experiments implicitly consider by using the re-analysis data. This could be tested by performing the same multivariate analysis of the solar cycle signal on the model run D_SFix; any statistically significant SCS signal derived from this must be due to dynamical feedbacks. My expectation would be that this looks very much like the other model experiments in the lower stratosphere, but shows no (significant) solar cycle signal in the upper stratosphere and mesosphere, where this is probably due to direct response of the chemistry. I would also not be surprised if the strong signal in the Southern high latitudes was a dynamical feedback (via vortex strength). So – please perform the same analysis on the model experiment D_SFix, and show / discuss results in Figures 3 and 4.

**We find that the both major comments are very insightful and they have helped us improve the manuscript significantly.**

To summarise, the two major concerns are:

1. Possible aliasing from 24 QBO terms
2. Implicit solar signal in the ERA5 forcing fields

**1. Aliasing effect of 24 QBO terms:** We have been using this approach since Dhomse et al., (2006). Basically, independent monthly terms means that the regression model has more degrees of freedom and the QBO terms are allowed to fit individual months. Two QBO terms help to capture the effects of both the speed and magnitude of the downward-propagating QBO. It also avoids aliasing effects compared to the other approaches such as using annual, semi-annual harmonics whereby forcing regression coefficients have to follow seasonal effects. We agree that by using monthly mean ozone anomalies, QBO is the largest contributor to ozone variability.

Hence, in the revised manuscript we added an additional Figure R1 (as new Figure 4) to show results from the composite analysis (mean ozone difference for solar maximum minus minimum months) and correlation analysis (correlation between ozone anomalies and Mg-ii

index) for the mean ozone profiles between 20S-20N. Ozone anomalies for this analysis are calculated by removing the linear trend at individual grid points (simple least square fit). The new Figure 4 clearly shows that all the three analysis methods (composite, correlation, regression) are consistent with each other: i.e. largest positive SCS in the mid-upper stratosphere and minimum SCS in the lower stratosphere (where QBO is largest contributor to the ozone variability). Thus, in the revised paper the new analysis as well as updated Figures 4 and 5 (now Figures 5 and 6 in revised manuscript) clearly illustrate that the regression analysis is not impaired by inclusion of 24 QBO terms.

Also, instead of adding contribution lines from different explanatory variables in Figure 1 (in order to avoid a very complex figure), we decided to add supplementary material to include regression coefficients for three prominent explanatory variables (QBO30, QBO50 and Age of Air) with large dynamical influence. Regression coefficients shown in (new) Figures 5 (Figure R2 below) and 6 as well as Supplementary Figures S1 to S3 confirm that dynamical processes are the largest contributor to the ozone variability (especially in the lower and middle stratosphere). Regression coefficients from D_SFix also illustrate that in the lower stratosphere, the implicit solar cycle does play key role in controlling ozone variability and as pointed out by the reviewer in the revised manuscript we note that the large SCS in the Antarctic stratosphere is primarily of dynamical origin.

Also in Section 3, we do have a discussion about the use of three regularisation regression methods (Lasso, Ridge and Elastic-Net) that considerably reduce variance in the regression model as well as avoid possible aliasing effects.

Re: Lines 201-207 - As for small differences between in ozone time series (Figure 3), we also added ozone anomalies from D_SFix and E_DFix (a new simulation with fixed repeating dynamics). As expected, lines for distinct simulations at upper stratospheric levels almost overlap in the lower stratosphere, confirming the chemical solar cycle influence is almost negligible in the lower stratosphere. Again, it is confirmed in the (new) Figures 5 and 6 where D_SFix shows a significantly different SCS compared to A_NRL, B_SATIRE and C_SORCE simulated ozone profiles.

[Figure]

*Figure R1: (a) MLS ozone profile composite for solar maxima (n=40) and solar minima (n=51) months (black line). Percentage ozone differences for the tropics (20°S-20°N) between a model simulation with time-varying NRL2 solar flux (A_NRL2), fixed solar flux (D_SFix) and fixed dynamics (E_DFix). Simulations are shown with blue, cyan and orange lines, respectively. (b) Correlation coefficient between Mg-ii index and monthly mean ozone anomalies from MLS, A_NRL2, D_SFix and E_DFix.*

As per the Reviewer's suggestion, in the new Figure 5 (and supplementary Figures S1 to S3) we included stippling to show the regions where regression coefficients are smaller than 1-sigma standard deviation.

**2. Implicit solar signal in ERA5**: First, we agree with the reviewer that we should have included more analysis of D_SFix profiles to elucidate effects of the implicit solar cycle signal in stratospheric ozone. In the revised manuscript Figures 5 and 6 include SCS estimates for D_SFix, which confirm that in the mid-lower stratosphere the implicit solar signal is primarily of dynamical origin. As mentioned above, there is a new figure (Figure R1 above, Figure 4 in paper) that shows solar maximum/minimum composites. These confirm that the implicit solar cycle signal in ERA5 is not enough to simulate observed SCS, especially in the middle-upper stratosphere. The E_DFix ozone time series at different levels also show that at 100 and 31 hPa there is very little chemical solar signal which is consistent with the Reviewer's comments.

[Figure]

*Figure R2: Latitude-pressure cross-sections of solar regression coefficients (or Solar Cycle Signal per 100 solar flux unit) for MLS (top row) as well as model simulation A_NRL2 (second row), B_SATIRE (third row), C_SORCE (fourth row) and D_SFix (bottom row). Regression coefficients are from OLS (first column), Lasso (second column), Ridge (third column) and Elastic Net (fourth column) regression models. Stippling indicate regions where regression coefficients are smaller than 1-sigma standard deviation.*

**Minor points:**

Line 16-19: "compared to earlier estimates" – I don't disagree with this sentence, but struggled with it nevertheless. I think the important differences are, on the one hand, much denser sampling of the observations by MLS independent of solar illumination, that is, also covering high; on the other hand, observations during a different time-period with (possibly) simpler background conditions leading to less (obvious) aliasing with other terms of variability.

**We think the sentence conveys the same message. We have now modified it slightly to say: "Finally, we argue that the overall significantly different SCS compared to earlier estimates might be due to a combination of different factors such as much denser MLS measurements, almost linear stratospheric chlorine loading changes over the analysis period, variations in stratospheric dynamics as well as relatively unperturbed stratospheric aerosol layer leading to less aliasing effects"**

Lines 155-156, Figure 1: you could provide the correlation coefficient as a measure of the quality of the fit. As the multivariate regression is essentially a multi-linear regression, Pearson's correlation coefficient is well suited for that.

**Added.**

Line 163, Figure 2: what is the meaning of the error bars? Are the derived from the error covariances of the multivariate regression, or from the variance within the sample, or both? Please explain.

**Clarifed as they are from "error covariance matrix".**

Line 166-167: You could / should discuss this around the error bars you provide for the MLS based SCS: In 35-45 km, the MLS-based SCS are significantly higher than the previous results, with the results from previous estimates outside the error range of the MLS-based SCS; the best agreement is observed for HALOE SCS, which is just at the edge of the lower error bound of the MLS SCS. In 50-56 km, MLS SCS and HALOE SCS agree within error bounds, but are significantly lower than the SAGE II SCS. In 20-30 km, all data-sets yield consistent results within the error bounds of the MLS SCS.

**Yes, we have revised the paragraph.**

Line 167 – 168: "A key feature is that the MLS SCS … is almost twice as large as any other satellite-data based SCS reported in the past" … considering that this was derived over a comparatively weak solar maximum, this result is somewhat surprising.

**Indeed. We have added bit more discussion in the revised manuscript. It is possible that this could be because of multiple factors. Analysis of D_SFix suggests that almost 30% is of dynamical origin. Other factors include that SSI variations among various wavelengths is not systematic and ozone production is largely controlled by irradiances below 240 nm whereas irradiances at longer wavelengths determine ozone loss.**

Line 173-174: Additionally to the sparse sampling, the solar occultation instruments only measure during a very specific time of day, sunrise and sunset, while MLS measures independent of solar illumination. This is probably not important in the lower and mid-stratosphere as ozone does not have a significant diurnal cycle there; but could it affect results in the mesosphere and uppermost stratosphere, where a diurnal cycle evolves?

**That is true. Here we primarily focus on stratospheric altitudes so effects of the diurnal cycle should be minimum (e.g. Dhomse et al, 2013, Figure 2).**

Line 185-187: the enhanced stratospheric aerosol also leads to lower ozone values, which will have an impact on the regression results I guess.

**We have added "Mt Pinatubo also caused significant ozone losses and changes in the stratospheric circulation".**

Line 188-190: agreed that SSI changes could have been different from earlier solar cycles, but would you expect a larger amplitude of the solar cycle signal for the weaker cycle?

**We do not know (see above). Now we have added a sentence to expand a bit more on this: "However, it does not mean that SCS during previous cycles would be larger as complications also arise from various complex couplings such as temperature feedback (increased direct radiative heating during solar maxima), wavelength-dependent photolysis rates (irradiance changes are not uniform across different wavelengths)".**

Line 198: "is most probably due to …" I would formulate this a bit more carefully. Maybe "is likely due to …"?

**Thanks, we have replaced "probably" with "likely".**

Line 200: why not do the same regression analysis as well for D_SFix? This would enable you to separate the purely chemical solar cycle signal from the implicit dynamical feedback contained in the use of re-analysis data.

**Good idea. In the revised manuscript we analyse D_SFix as well as a new simulation (E_DFix) where dynamics is fixed.**

Line 218-219: for instruments depending on solar illumination, high latitudes are naturally difficult, in particular as they would certainly miss polar night.

**Indeed, that issue is highlighted in point 1 above.**

Lines 223 – 225: again, how are the error bars derived? And – all model results seem to agree with MLS within error bars over the whole range shown (though that is difficult to assess in the figure), so the differences should probably not be overinterpreted.

**Yes, we are keen not to over-interpret the results (see discussion about D_SFix results). Also see replies to the earlier comments.**

Figure 6 – can you change "plev" on the y-axis to "pressure (hPa)"?

**Done.**

Line 251: … significant variations "of the" ozone difference patterns …

**Modified as "significant variations in the ozone difference patterns".**

Line 264-265: just as a suggestion for the future – would it be possible to include other instruments measuring independent of solar illumination into the analysis, e.g., MLS/UARS, MIPAS/ENVISAT, SMR/ODIN?

**We thank the reviewer for useful suggestion for the future. Adding data from other instruments (especially SMR/ODIN) could strengthen our arguments. Clearly, due to time constraints we are not able to do so in this manuscript, but in future studies we would surely aim to include data from other instruments.**